# Utilizing NF-κB Signaling in Porcine Epithelial Cells to Identify a Plant-Based Additive for the Development of a Porcine Epidemic Diarrhea Virus Vaccine

**DOI:** 10.3390/vetsci12020181

**Published:** 2025-02-18

**Authors:** Nguyen-Thanh Hoa, Haroon Afzal, Thu-Dung Doan, Asad Murtaza, Chia-Hung Yen, Yao-Chi Chung

**Affiliations:** 1International Program in Animal Vaccine Technology, National Pingtung University Science and Technology, Pingtung 91201, Taiwan; 2Department of Virology, National Institute of Veterinary Research NIVR, Hanoi 11500, Vietnam; 3Graduate Institute of Animal Vaccine Technology, College of Veterinary Medicine, National Pingtung University of Science and Technology, Pingtung 91201, Taiwan; 4Faculty of Biosciences, Fisheries and Economics, Norwegian College of Fishery Science, UiT—The Arctic University of Norway, P.O. Box 6050 Tromsø, Norway; 5Graduate Institute of Natural Products, College of Pharmacy, Kaohsiung Medical University, Kaohsiung 807378, Taiwan

**Keywords:** NF-κB signaling pathway, porcine epidemic diarrhea virus (PEDV), epithelial cells, PK15-KBR cell line, plant extracts, in vitro tool, mucosal immunity

## Abstract

A porcine epithelial NF-κB reporter cell line was created to identify plant extracts that activate the NF-κB signaling pathway. Among the 224 screened extracts, 3 showed an increase in NF-κB activity and were further assessed for their potential application as additives in PEDV vaccines. Toxicity assessments, both in vitro and in vivo, were performed, resulting in the selection of chamomile as a candidate for an in vivo vaccination study. Mice received inactivated PEDV vaccinations, formulated with a commercial adjuvant and with or without the addition of chamomile, via oral and intramuscular (I.M.) routes. Vaccine efficacy was determined by measuring virus-specific IgG and IgA antibodies. Chamomile did not affect the I.M. vaccination but showed a propensity to elevate antibody titers for the oral vaccination. These results indicate that chamomile extract is worthy of more study as a potential ingredient in PEDV vaccines. This study underscores the potential of PK15-KBR cells in identifying potential mucosal adjuvants and positions chamomile extract as a valuable candidate for enhancing the creation of effective vaccines for PEDV alongside potential other viral diseases in animals.

## 1. Introduction

NF-κB plays a crucial role in maintaining immune homeostasis and regulating inflammation in epithelial tissues, which are primary targets for enteric viruses [1]. One significant pathogen affecting these tissues is the porcine epidemic diarrhea virus (PEDV), which is responsible for severe watery diarrhea and high mortality rates in neonatal piglets, making it a major cause of diarrhea outbreaks on pig farms worldwide [2]. Epithelial cells serve as the first line of defense against such infections by establishing a protective monolayer, secreting chemical barriers, and activating immune processes. These activities are integral to the innate immune system’s role in protecting the mucosal barrier [3]. Upon detecting pathogens like PEDV, NF-κB is activated, and multiple aspects of immune response are triggered through the production of antimicrobial peptides (AMPs) and other effector molecules [4]. This mechanism is essential for an effective immune response and maintaining the integrity of epithelial barriers.

Several plant extracts have been shown to significantly modulate NF-κB signaling pathways in epithelial cells, highlighting their potential therapeutic roles in managing inflammation and related diseases [5,6]. They hold great potential as vaccine adjuvants due to their safety, long-lasting immunity, dosage sparing, and economic benefits [7]. Additionally, oral administration of plant-derived adjuvants has been shown to reduce the duration of clinical symptoms, decrease mortality risks, and improve overall outcomes compared to control treatments [8].

Unlike in vivo experiments, cell-based assays provide more direct, verified, and high-throughput platforms for screening novel adjuvants [9]. This approach allows for evaluating a large library of plant extracts and isolates to discover novel modulators of NF-κB activity. Although the adjuvant activities have been successfully investigated in immune cells, there are limitations in evaluating these plant products on epithelial cells [10,11]. Given their importance, epithelial cells represent promising in vitro systems for identifying new mucosal adjuvants derived from natural products, such as plant extracts.

Despite the use of NF-κB reporter cells to evaluate immunomodulatory products, most platforms rely on immune cells, such as T cells [12] and B cells [13], or antigen-presenting cells, including macrophages [14] and dendritic cells [10]. The source of the cell lines used in these assays is important, as porcine-derived epithelial cell lines can offer a more accurate model for assessing the innate responses in pigs. Previous research has focused on viral or bacterial stimuli, such as transmissible gastroenteritis virus (TGEV) and *Haemophilus parasuis*, to study NF-ᴋB signaling in PK-15 cells [15,16]. The impact of plant-derived adjuvants on NF-κB signaling in PK-15 cells highlights a critical gap in the research.

To address this gap, we developed a stable NF-κB-luciferase cell line derived from the PK-15 porcine cells, referred to as PK15-KBR. This cell line maintained stability for at least ten passages in culture, retaining robust reporter gene expression. Upon stimulation with TNF-α, PK15-KBR exhibited clear NF-κB activation, as confirmed by luciferase assays. The platform was further tested on a 96-well plate, indicating its suitability for high-throughput screening. Using this system, we conducted a preliminary screening of crude plant extracts, followed by toxicity assessments and vaccination studies in mice. This comprehensive approach provides a valuable application for identifying and selecting potential mucosal adjuvants.

## 2. Materials and Methods

### 2.1. Cells and Viruses

Porcine kidney cells (PK-15, BCRC #60057) and African green monkey kidney cells (Vero cells, BCRC #60013) were obtained from the Bioresource Collection and Research Center (BCRC), Taiwan. Both cell lines were cultured in Minimum Essential Medium Alpha Medium (αMEM, Gibco, New York, NY, USA), supplemented with 5% heat-inactivated fetal bovine serum (FBS, Gibco, USA), in a humidity-controlled incubator at 37 °C with 5% CO_2_. Cell lines were monitored for viability and sterility throughout the study period.

PED virus was isolated from infected pigs and propagated in Vero cells, as described previously [17]. Briefly, cells were inoculated with PEDV in the infected media αMEM supplemented with 5 µg/mL of Trypsin-TPCK (Sigma, Carlsbad, CA, USA) and 1% *v*/*v* Penicillin-Streptomycin (Gibco, USA) in a humidity-controlled incubator at 37 °C with 5% CO_2_. The cytopathic effects (CPE) were monitored daily to record clear signs of cell rounding, fusion, and detachment. After 72 h after infection, infected cells were lysed using the freeze–thaw method and centrifuged at 2000× *g* for 10 min. Virus titer in the supernatant was determined using a 50% tissue culture infectious dose (TCID_50_) assay [18]. The viral suspension was adjusted to a final concentration of 10^7^ TCID_50_/mL and heat-inactivated at 56 °C for 30 min. The inactivated virus was aliquoted and stored at −80 °C until use.

### 2.2. Establishment of Reporter Cell Line

To monitor NF-κB activation, a porcine kidney epithelial cell line (PK-15) expressing an NF-κB reporter gene was generated as follows: PK-15 cells were transduced with a lentiviral vector encoding the NF-κB consensus binding sequence, luciferase, and enhanced green fluorescent protein (eGFP). The lentivirus plasmid, pHAGE NFkB-TA-LUC-UBC-GFP-W, was a gift from Darrell Kotton (Addgene plasmid # 49343; http://n2t.net/addgene:49343; RRID: Addgene_49343, accessed on 12 February 2013). This dual reporter system enables independent tracking of successfully transduced cells via eGFP fluorescence and allows for real-time assessment of NF-κB activation through luciferase expression (Figure 1A).

PK-15 cells were incubated overnight with lentivirus-containing supernatant in the presence of 5 µg/mL polybrene (Merck, Darmstadt, Germany) to enhance transduction efficiency. Subsequently, cells were harvested and analyzed for eGFP expression using flow cytometry (BD FACScan, BD Biosciences, Palo Alto, CA, USA) with data processed in FlowJo software version 10.8.1 (BD Biosciences, Palo Alto, CA, USA). Single eGFP-positive cells were sorted into individual tubes. The successfully transduced cell line, referred to as PK15-KBR, was selected and passaged for further studies.

### 2.3. Flow Cytometry

Flow cytometry was utilized to assess the transduction efficiency of PK15-KBR cells, specifically measuring the percentage of GFP-positive cells. Briefly, 5 µL of 7-Amino-Actinomycin D (7-AAD) (BD Biosciences, Palo Alto, CA, USA) was added to get rid of dead cells. Flow cytometry was conducted using a BD FACScan instrument (BD Biosciences, USA), while cell sorting experiments were performed on a BD FACSMelody™ 4-Way Cell Sorter (BD Biosciences, Palo Alto, CA, USA). The data analysis was completed using FlowJo software version 10.8.1 (BD Biosciences, Palo Alto, CA, USA) to quantify transduction efficiency and evaluate cell population.

### 2.4. Preparation of Plant Extracts (PEs)

A total of 224 crude extracts (PEs) were obtained from TFDA food ingredients reference plant extract library (TFDA-RD, Natural Product Libraries and High-throughput Screening Score, Kaohsiung Medical University). Whole plant sources from Taiwan were processed into extracts following standard protocols. Dried plant materials were soaked in cold methanol at room temperature for three days, with the process repeated three times to maximize extraction efficiency. The resulting aqueous extracts were weighed and dissolved in culture-grade dimethyl sulfoxide (DMSO, Merck, Darmstadt, Germany) to achieve a final concentration of 100 mg/mL for subsequent use.

### 2.5. Quantification of NF-ᴋB Activation via TNF-α Stimulation

NF-ᴋB activation was quantified by measuring luciferase activity in TNF-α stimulated PK15-KBR cells using the Luciferase Assay System (Promega Corporation, Madison, WI, USA). Briefly, cells were seeded overnight on 96-well plates at 37 °C with 5% CO_2_ to allow monolayer formation. Reporter cells were then incubated with tumor necrosis factor-alpha (TNF-α) at a concentration ranging from 0.625 to 10 ng/mL alongside DMSO controls at concentrations of 0 to 1% (*v*/*v*). Cells were incubated at 37 °C with 5% CO_2_ for 16 h. After incubation, the medium was removed, and cells were lysed with 50 μL of 1 × passive lysis buffer for 15 min at room temperature. Finally, 20 μL of the lysate was mixed with 50 μL of Luciferase Buffer for luciferase activity measurement, while another 20 μL of lysate was mixed with 50 μL of Alamar Blue for cell viability assessment. Both luciferase activity and cell viability were measured using the GloMax-Multi+ Detection System (Promega Corporation, Madison, WI, USA).

### 2.6. High-Throughput Screening Assays

High-throughput screening was performed to evaluate NF-κB signaling in PK15-KBR cells in response to plant extracts (PEs). A monolayer of 10,000 cells per well was treated with PEs at a final concentration of 10 µg/mL. TNF-α was used as the positive control at 10 ng/mL, and DMSO served as the solvent control at 0.1% (*v*/*v*). Cells were inoculated at 37 °C with 5% CO_2_ for 16 h. After incubation, the medium was removed, the cells were lysed with 50 μL of 1 × passive lysis buffer for 15 min at room temperature. Finally, 20 μL of the lysate was mixed with 50 μL of Luciferase Buffer for luciferase activity measurement, and another 20 μL of lysate was mixed with 50 μL of Alamar Blue to assess cell viability. Both luciferase activity and cell viability were measured using GloMax-Multi+ Detection System (Promega Corporation, Madison, WI, USA).

### 2.7. CCK-8 Assays

Cytotoxicity was evaluated using the colorimetric Cell Counting Kit-8 (CCK-8, Abcam, Taipei, Taiwan), a sensitive assay for quantifying cell viability. A monolayer of 5000 PK-15 cells was pre-incubated in a 96-well plate overnight in a humidified incubator at 37 °C with 5% CO_2_. The following day, 10 μL of PEs at different concentrations (1—1000 µg/mL) was added to wells. DMSO at 0.01% *v*/*v* served as the control group, while αMEM medium was used as a background. The cells were incubated at 37 °C for 24 h. Finally, 10 μL of CCK-8 solution was added to each well and incubated at 37 °C for 3 h. Absorbance values at 450 nm were measured using a microplate reader. Cytotoxicity was calculated using the following formula:$$Cytotoxicity\;(\%)=\left(\;\frac{Absorbance\;of\;control\;cells-Absorbance\;of\;treated\;cells}{Absorbance\;of\;control\;cells-Absorbance\;of\;background}\;\right)\;\times 100$$

### 2.8. Toxicity Assessment in Mice Study

The toxic effects of intramuscular injections of PEs were evaluated using a modified methodology in a study involving 25 seven-week-old ICR mice. The mice were randomly divided into five groups (*n* = 5) as follows: (1) 25 mg/kg body weight (bw) of chamomile extract (Chamomile_Low), (2) 50 mg/kg bw of chamomile extract (Chamomile_High), (3) 25 mg/kg bw of mulberry extract (Mulberry_Low), (4) 50 mg/kg bw of mulberry extract (Mulberry_High), and (5) phosphate-buffered saline (PBS) as the control group (Table 1) [19]. Each mouse received 100 µL of the assigned treatment via intramuscular injection. The concentration of DMSO was maintained below the maximum permissible concentration of 10% in all control and treatment group samples. Signs of toxicity, including adverse symptoms and mortality, were closely monitored and recorded during the first 24 h post-injection.

### 2.9. Experimental Design and Sample Collection

For intramuscular injection, seven-week-old female ICR mice were obtained from BioLASCO Taiwan Co., Ltd. (Taipei, Taiwan) and randomly assigned to four groups (*n* = 5) as follows: (1) in-house inactivated PEDV (IM), (2) in-house inactivated PEDV with 6 mg/kg bw of chamomile extract (IM 120), (3) in-house inactivated PEDV with 24 mg/kg bw of chamomile extract (IM 480), and (4) PBS as the control (Table 2). Each vaccine was formulated with the water-in-oil adjuvant MONTANIDE™ ISA 206 VG (Seppic, Castres, France) in a 1:1 ratio. Each mouse received intramuscular injections of 100 µL on days 0 and 14, following the protocol established in our previous study [20].

For oral administration, seven-week-old female ICR mice were obtained from BioLASCO Taiwan Co., Ltd. and divided into four groups (*n* = 5) as follows: (1) in-house inactivated PEDV (OR), (2) in-house inactivated PEDV with 6 mg/kg bw of chamomile extract (OR 120), (3) in-house inactivated PEDV with 24 mg/kg bw of chamomile extract (OR 480), and (4) PBS as the control (Table 2). Vaccines for oral administration were formulated with the water-in-oil adjuvant MONTANIDE™ GR01 (Seppic, France) in a 7:3 ratio. Mice were administered 100 µL doses orally once per week for four consecutive weeks [21].

Serum samples were collected weekly, while the intestinal fluid samples were obtained on day 28 for subsequent immune evaluation.

### 2.10. Indirect Enzyme-Linked Immunosorbent Assay (ELISA)

ELISA were conducted to qualify total IgG antibodies specific to PEDV in mouse sera. Briefly, the sera were separated from whole blood by centrifugation at 800× *g* for 5 min and stored at −20 °C until use. In-house inactivated PEDV was coated onto 96-well plates at a concentration of 100 TCID_50_ per well and incubated at 4 °C overnight. The following day, plates were blocked with 3% BSA at 37 °C in one hour. After washing with PBST, four-fold diluted sera were added into the wells and incubated at 37 °C for 1.5 h as a primary antibody. Subsequently, plates were incubated with anti-mouse IgG or IgA antibody-peroxidase conjugate (Sigma, Carlsbad, CA, USA) diluted to 1:10,000 as a secondary antibody at 37 °C for one hour. After the final wash with PBST, TMB substrate solution was added for color development. Finally, TMB Stop Solution was applied to stop the reaction. The color intensity was quantified at a wavelength of 450 nm using a microplate reader, reflecting the relative antibody levels.

### 2.11. Statistical Methods

NF-κB determination and cytotoxicity tests were repeated independently at least three times to ensure data reproducibility. Statistical analyses and graphical presentations were performed using JASP version 0.19.3.0 JASP software (NIEUW-VENNEP, Netherlands) and GraphPad Prism version 10 (GraphPad Software, San Diego, CA, USA). Non-parametric Kruskal–Wallis tests were conducted for cytotoxicity assays, NF-κB quantification, and determination of IgA antibody titers. An independent sample *t*-test was used to determine the difference between two samples in this group. Two-way ANOVA with Dunnett’s post hoc test was applied to evaluate IgG antibody titers, followed by GLMMs for each of the models. Data are presented as the mean of the triplicate of antibody titer per subject with ± SEM. Statistical significance was denoted as follows: *p* ≤ 0.05 (*), *p* ≤ 0.01(**), *p* ≤ 0.001 (***), and *p* ≤ 0.0001 (****).

## 3. Results

### 3.1. Establishment of PK15-KBR Cells

To facilitate real-time tracking of NF-κB activation, we developed a reporter cell platform, PK15-KBR cells, incorporating the lenti-NF-κB-luc-GFP plasmid. Transduction efficiency was evaluated by measuring the mean fluorescence intensity of GFP-positive cells through flow cytometry. Initial analysis revealed that approximately 4% of the infected PK-15 cells expressed green fluorescence, with a dramatic increase to over 95% after cell sorting. This high level of transduction efficiency was maintained stable for up to ten passages. However, a marked reduction in transduction proficiency was observed after freeze–thaw cycles, suggesting the necessity of repeated cell sorting to maintain a sufficient population of transduced cells for downstream applications. These observations are summarized in Figure 2.

### 3.2. NF-κB Activation by TNF-α Stimulation in a Dose-Dependent Manner

To validate the functionality of the lenti-NF-κ B-luc-GFP construct in PK15-KBR cells, we conducted preliminary testing using TNF-α, a well-known activator of NF-κB signaling (Figure 3A). PK15-KBR cells were exposed to TNF-α at concentrations ranging from 0.625 to 10 ng/mL. Luciferase activity in PK15-KBR cells showed a significant dose-dependent increase compared to the untreated control group. Luciferase activity was three-fold higher than the control at the lowest concentration (0.625 ng/mL), after which it experienced a rise to seven-fold at highest concentration (*p* < 0.001) (a detailed analysis is provided in Appendix A). These results demonstrate that TNF-α effectively induces NF-κB-driven luciferase activities in PK15-KBR cells in a dose-dependent manner.

Further analysis revealed that the addition of 1% DMSO impacted luciferase responses under TNF-α stimulation (Figure 3C). A consistent increase in luciferase activity was observed across all TNF-α concentrations (0.625 to 10 ng/mL), regardless of DMSO presence (Figure 3, panel A and B). However, cells treated with 1% DMSO exhibited a notable reduction in overall luciferase activity compared to the cells stimulated without DMSO. At lower TNF-α concentrations (0.625 to 1.25 ng/mL), NF-κB-driven luminescence increased less than two-fold, whereas at higher concentrations, the increase was approximately three-fold with statistical significance (a detailed analysis is provided in Appendix A).

In the final trial, luminescence responses were systematically evaluated under different experimental conditions, including cell numbers, drug action time, and solvent concentrations. TNF-α at 10 ng/mL was defined as the positive control, and DMSO served as the solvent control (Figure 3C). Stimulation with TNF-α resulted in a significant increase in luminescence compared to the solvent control group (*p* < 0.001) (a detailed analysis is provided in Appendix A). To assess the robustness of the assay for high-throughput screening, the Z’ factor was calculated from luminescence values of positive and negative controls. The Z’ value of 0.68 confirmed the suitability of the assay for high-throughput applications.

Collectively, these findings confirm that TNF-α induces NF-κ B-driven luciferase activity in PK15-KBR cells in a dose-dependent manner, with or without the presence of DMSO, although DMSO may suppress the magnitude of the luminescence response.

### 3.3. NF-κB Activation by Chamomile Extract in a Dose-Dependent Manner

A total of 224 crude plant extracts from the TFDA-RD library were screened for their ability to activate NF-κB signaling. Among the tested extracts, chamomile, *Boerhaavia diffusa* and mulberry demonstrated NF-κB activation, while the remaining extracts produced background luminescence levels comparable to the solvent control group.

Subsequent experiments confirmed the NF-κB activation potential of these three extracts across five different concentrations. Chamomile exhibited clear dose-dependent activation of NF-κB, while *Boerhaavia diffusa* and mulberry showed partial dose dependence compared to the control group (Figure 4). These results highlight chamomile extract as the most promising candidate among the screened PEs, with its ability to induce NF-κB activation potentially supporting its role in eliciting an effective immune response.

### 3.4. Toxicity Assessment of PEs In Vitro and In Vivo

In vitro cytotoxicity is a useful method for evaluating biological materials due to its simplicity, speed, and high sensitivity [22]. Overall, there was no statistical difference between the treatment groups, but there was a clear difference in different concentrations. Of the three screened PEs, mulberry and boerhavia diffusa exhibited mild cytotoxicity at 1000 µg/mL (*p* < 0.5) (a detailed analysis is provided in Appendix A). Most results were well below the 30% cytotoxicity threshold, meeting the standard outlined in ISO 10993-5 [23]. At tested concentrations of 10, 100, and 1000 µg/mL, all three extracts showed no significant cytotoxic effects compared to control groups (Figure 5).

The potential in vivo toxicity of chamomile and mulberry was further evaluated in a mouse model (Figure 6). Mice treated with these extracts displayed no abnormal behaviors or adverse clinical signs across all experimental groups during the observation period. These findings confirmed that both chamomile and mulberry were safe for further experimentation.

### 3.5. PEDV-Specific Antibody Titers Increased by Chamomile-Adjuvanted and In-House Inactivated PED Vaccines

To assess the impact of PEs on immunogenicity and to bridge the in vitro NF-κB activation with in vivo immune responses, chamomile was selected for further evaluation due to its distinct in vitro effects. Serum IgG antibody responses were measured in vaccinated mice using an indirect ELISA with in-house inactivated PEDV as the coating antigen (Figure 7).

Mice administered via injection demonstrated markedly higher serum antibody titers at all time intervals in comparison to those receiving the oral vaccination. In both vaccination methods, antibody responses increased in a time-dependent manner, with the lowest titers recorded on day 7 and the highest on day 28, indicating the immune system’s response to the vaccine matures over time. Significant differences were seen among the oral vaccine groups on day 14. Mice administered the chamomile-adjuvanted PEDV vaccination exhibited substantially higher IgG antibody levels compared to those in the inactivated (OR) group (OR 120, *p* < 0.01; OR 480, *p* < 0.05). In contrast, the intramuscular (I.M.) vaccinations did not show any significant effect from the presence of chamomile on IgG antibody titers (detailed analysis is provided in Appendix A). This indicates that while chamomile may enhance oral vaccine formulations, its impact is not observed in injectable forms. A key finding from this study was that the addition of chamomile to the vaccine formulation enhanced IgG antibody responses in serum, starting from day 14 post-vaccination in the oral vaccination groups. This finding underscores the potential of natural adjuvants in vaccine development and emphasizes the need for tailored approaches based on vaccination routes to optimize immune responses.

### 3.6. IgA Antibody Enhancement in Mice Vaccinated with Chamomile as an Adjuvant

IgA antibody responses were evaluated in intestinal fluid samples collected on day 28 following the first vaccination, using indirect ELISA with in-house inactivated PEDV as the coated antigen (Figure 8). The OR 480 group of mice exhibited the highest IgA antibody titers compared to the control group, although this difference was not statistically significant. Similarly, no significant effect from the addition of chamomile on IgA titers was observed across the injection groups (a detailed analysis is provided in Appendix A). These findings suggest a potential trend indicating that chamomile may enhance IgA production in mice and underscore the need for further investigation into its mechanisms of action and implications for vaccine development.

## 4. Discussion

Vaccines delivered through the mucosa can efficiently boost both innate and acquired immune responses [3,24]. Traditional animal-based studies can be reliable but time-consuming and costly. To address these limitations, high-throughput assays have emerged as efficient alternatives for predicting the potential of novel materials. Combining in vitro assays with animal models allows for a more comprehensive and accurate assessment of novel adjuvanticity by leveraging the strengths of both methods [25]. An ideal predictive method should be highly accurate, fast, reliable, cost-effective, and require minimal sample quantities. Our study incorporates these principles to refine the process of adjuvant selection.

The activation of NF-κB signaling in PK15-KBR cells served as a robust platform for cellular screening in this study. Among the three screened plant extracts, chamomile upregulated significant NF-κB activity in vitro and enhanced humoral immune responses in vivo. Previous studies have highlighted the predictive value of in vitro NF-κB activation in correlating with in vivo immune responses, establishing its reliability as a model system. For instance, TLR reporter cell lines, such as those from Invitrogen, have been widely used to assess TLR activation and its downstream signaling, including NF-κB activity, which strongly correlates with immune activation in vivo [26]. Consistent with these findings, our study showed a dose-dependent activation of NF-κB in PK15-KBR cells following chamomile treatment, which corresponded with a dose-dependent increase in IgA antibody titers when used as an oral adjuvant with an inactivated PEDV vaccine. These results highlight the predictive link between in vitro NF-κB activation and in vivo immune responses, emphasizing chamomile’s potential as an immunomodulator in vaccine formulations.

In vitro studies commonly use mucosa-associated cell lines to explore activities related to adjuvanticity. Immortalized cell lines, such as the epithelial cells TC-1, dendritic cells Jaws II, and macrophage Raw 264.7, have been widely used due to their reliability and practicality [25]. Among these, epithelial cells are particularly advantageous for the high-throughput screening of materials that impact mucosa function [27]. While primary intestine cell-derived models may offer greater physiological relevance for evaluating mucosal adjuvants for PEDV vaccinations, their cultivation complexity limits broader applicability [9]. The PK-15 porcine epithelial cell line provides a suitable alternative due to its broad expression of immunological receptors, making it relevant in various bioactivity assays [28]. The dual reporter system in PK15-KBR cells offers distinct advantages: GFP intracellular expression facilitates the sorting of positively transduced cells using flow cytometry, while luciferase secretion enables quick and efficient readouts. Moreover, NF-ᴋB-based reporter cell lines, such as PK15-KBR, have been instrumental in identifying new immunomodulatory compounds and revealing cytokine expression mechanisms [13]. This aligns with the critical role of NF-ᴋB signaling in epithelial cells for immune regulation [4].

Recent research underscores the immunomodulatory potential of natural products, including plant-derived compounds, in the development of novel mucosal adjuvants [29]. Our study reveals a novel role of chamomile as an immunomodulator, enhancing NF-κB activation and antibody responses when used as a supplement with an orally inactivated PEDV vaccine. Unlike earlier studies that emphasized chamomile’s ability to suppress NF-κB activity via IκB kinase inhibition and reduced RelA/p65 nuclear translocation in murine macrophages [30], our findings demonstrate a dose-dependent activation of NF-κB in PK15-KBR cells. This outcome underscores the complexity of chamomile’s immune interactions, which may vary depending on dose, formulation, or cellular context. Moreover, the oral administration of chamomile may enhance IgA production in certain treatments or conditions, in contrast to intramuscular administration, which showed no notable improvement in immune response. This observation suggests that chamomile’s immunostimulatory effects are enhanced by interaction with gut-associated lymphoid tissues (GALT), highlighting its potential as a strategic oral adjuvant.

Despite its promising findings, this study has limitations and warrants further research to refine the PK15-KBR reporter cell system and explore the mechanisms underlying chamomile’s immunomodulatory effects. The precise immune activation pathways remain unclear. Previous studies have focused on chamomile’s anti-inflammatory and antioxidant properties, attributed to bioactive constituents such as apigenin, bisabolol, and polysaccharides [30,31]. Future studies should isolate and evaluate the contributions of these compounds to NF-κB activation in PK-15 cells and elucidate the mechanisms underlying this activity. NF-κB activation involves two major signaling pathways: the canonical pathway, which responds to diverse stimuli, and the noncanonical pathway, which selectively reacts to specific ligands from the TNFR superfamily [4]. Further research is essential to understand how these pathways interact and regulate immune responses, tissue integrity, and inflammation in the context of chamomile-based immunomodulation. In addition, while the OR 480 group’s higher IgA levels are notable, further studies in larger animals are necessary to clarify aspects of the immune response beyond IgA production, such as T cell activation and cytokine profiles. This could provide a more comprehensive understanding of its role in mucosal immunity.

## 5. Conclusions

This study highlights the potential of the PK15-KBR reporter cell line as a robust, high-throughput platform for screening novel epithelial cell-targeted adjuvants. Among the tested extracts, three effectively induced NF-κB activation, with the chamomile extract demonstrating concentration-dependent activation in vitro and enhanced immune responses in mice. These findings establish PK15-KBR as a promising tool for identifying mucosal adjuvants, paving the way for future advancements in mucosal vaccine development.

## Figures and Tables

**Figure 1 vetsci-12-00181-f001:**
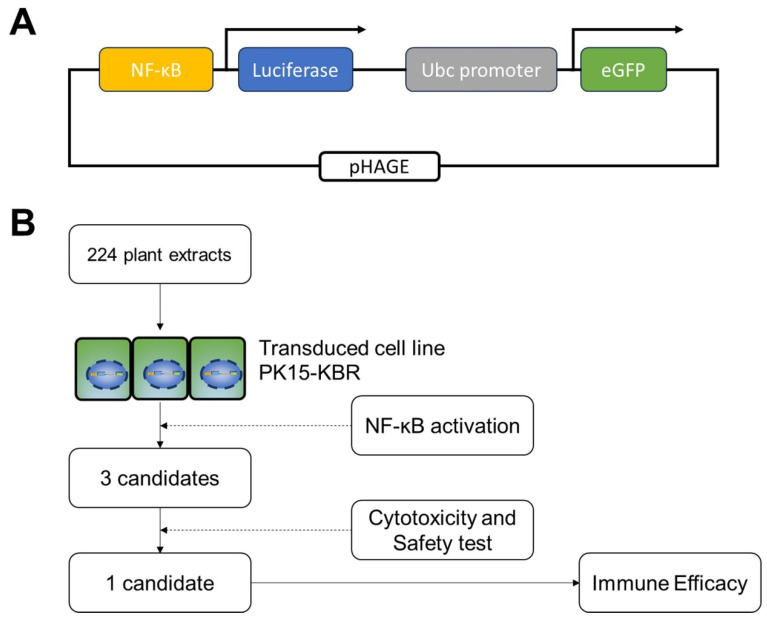
The concept of PK-15 reporter cell line based on NF-κB activation screening. (**A**) Schematic of transfected plasmid pHAGE-NF-κB-luc-GFP (Plasmid #49343). (**B**) The workflow for the screening of plant extracts using NF-κB driven luciferase assay, toxicity, and immune efficacy analyses.

**Figure 2 vetsci-12-00181-f002:**
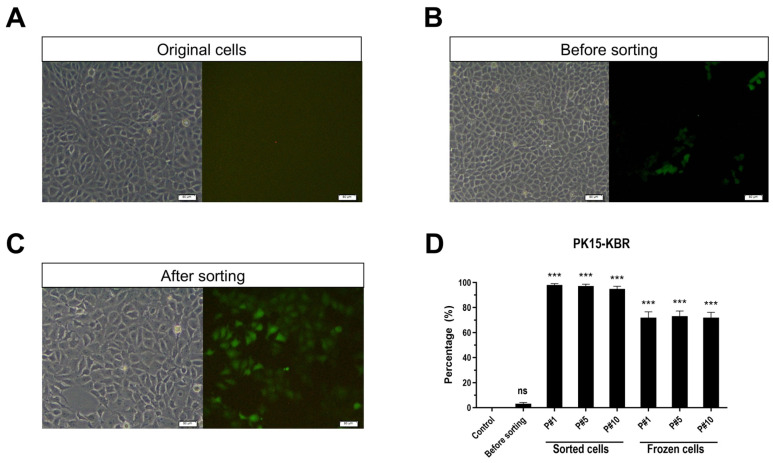
Morphology and fluorescence percentages of PK15-KBR cells. (**A**) The morphology of PK-15 cells without transduction by lenti-NF-κB-luc-GFP was captured using light and fluorescence microscopes at 10× magnification. (**B**) The morphology of transduced PK-15 cells that were GFP-positive was captured using light and fluorescence microscopes at 10× magnification. (**C**) The morphology of the sorted GFP-positive cells was captured using light and fluorescence microscopes at 10× magnification. (**D**) The percentages of GFP-positive PK-15 cells were measured using flow cytometry analysis at different time points. *** indicates significant differences (*p* ≤ 0.001), ns indicates non-significant.

**Figure 3 vetsci-12-00181-f003:**
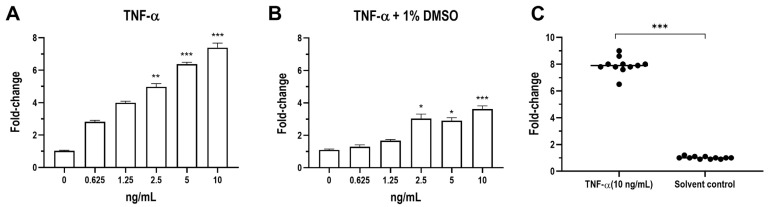
NF-ᴋB quantification of transduced PK15-KBR cells after stimulation with a series of TNF-α concentrations. Dose–response analysis of NF-ᴋB activity was measured using luminescence production via Luciferase assay system. (**A**) The fold-change of NF-ᴋB activity in PK15-KBR cells exposed to varying concentrations of TNF-α (ng/mL). (**B**) The fold-change of NF-ᴋB activity in PK15-KBR cells exposed to different conditions (cell numbers, drug action time, and concentration of solvent control). (**C**) The fold-change of NF-ᴋB activity in PK15-KBR cells exposed to varying concentrations of TNF-α (ng/mL) with addition of 1% DMSO as solvent control. The statistical analysis used in this study is detailed in Section 2.11. Significant differences are as follows: *p* ≤ 0.05 (*), *p* ≤ 0.01 (**), *p* ≤ 0.001 (***).

**Figure 4 vetsci-12-00181-f004:**
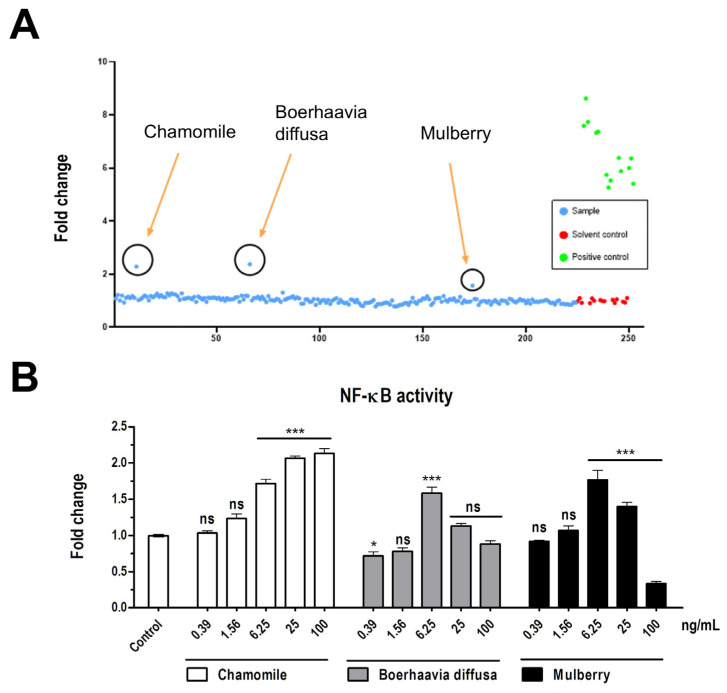
NF-ᴋB quantification of transduced PK15-KBR cells after stimulation with plant extracts. (**A**) The fold-change of NF-ᴋB activity in PK15-KBR cells exposed to plant extracts at a concentration of 10 μg/mL. DMSO (0.1%) was defined as solvent control and TNF-α (10 ng/mL as positive control. (**B**) The fold-change of NF-ᴋB activity in PK15-KBR cells exposed to varying concentrations of three extracts: chamomile, *Boerhaavia diffusa*, and mulberry. The statistical analysis used in this study is detailed in Section 2.11. Significant differences are as follows: *p* ≤ 0.05 (*), *p* ≤ 0.001 (***), ns (non-significant).

**Figure 5 vetsci-12-00181-f005:**
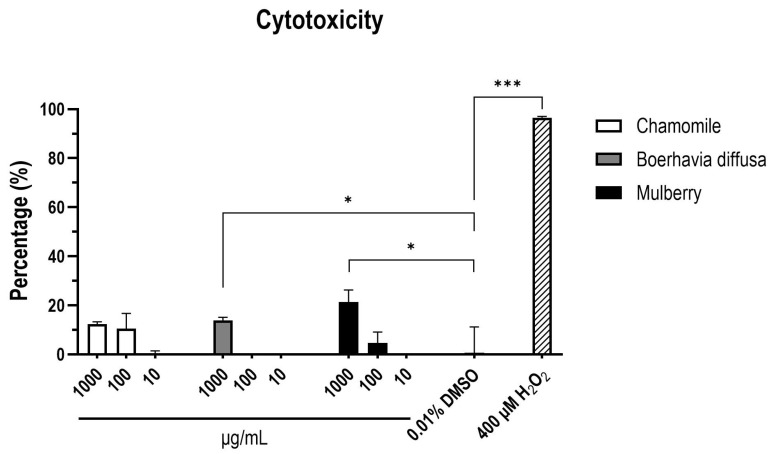
Cytotoxicity assay. CCK-8 assays were conducted to measure the cytotoxicity effects of PEs in PK-15 cells. The statistical analysis used in this study is detailed in Section 2.11. Significant differences are as follows: *p* ≤ 0.05 (*), *p* ≤ 0.001 (***).

**Figure 6 vetsci-12-00181-f006:**
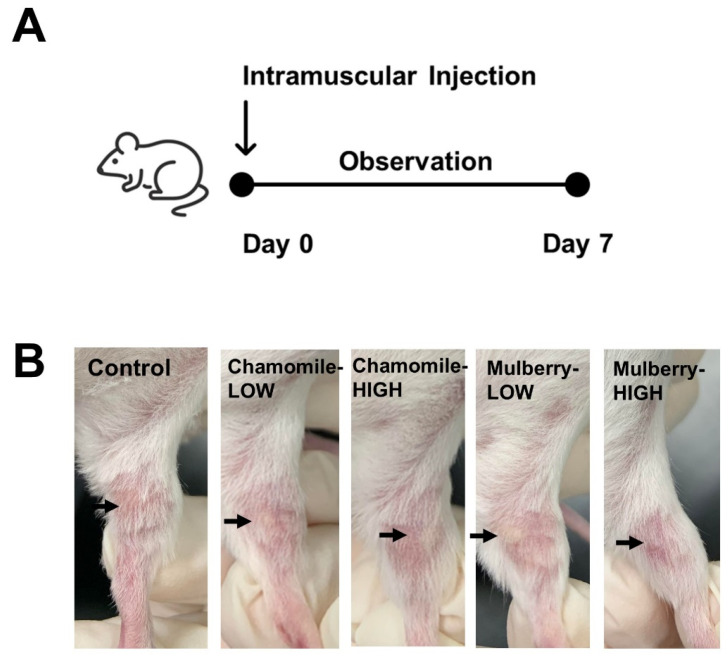
Safety test of intramuscular injection in mice. (**A**) Experimental design for mice study. (**B**) Observation of the injection site 7 days after the injection of each group of vaccines. The point of arrow is the point of injection.

**Figure 7 vetsci-12-00181-f007:**
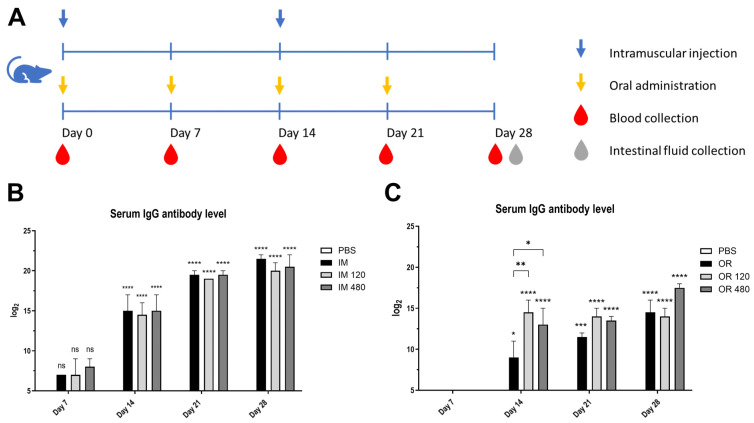
Serum IgG antibody level in mice vaccinated with in-house inactivated PEDV and chamomile extracts. Indirect ELISA assays were conducted to measure serum antibody titer using in-house inactivated PEDV as the antigen. (**A**) Experimental schedule. (**B**) Serum IgG antibody level in mice intramuscularly injected with different doses of chamomile extracts: IM (0 mg/kg bw), IM 120 (6 mg/kg bw), IM 480 (24 mg/kg bw), and PBS as the control group. (**C**) Serum IgG antibody level in mice orally fed with different doses of chamomile extracts: OR (0 mg/kg bw), OR 120 (6 mg/kg bw), OR 480 (24 mg/kg bw), and PBS as the control group. The statistical analysis used in this study is detailed in Section 2.11. Significant differences are as follows: *p* ≤ 0.05 (*), *p* ≤ 0.01(**), *p* ≤ 0.001 (***), *p* ≤ 0.0001 (****) and ns (non-significant).

**Figure 8 vetsci-12-00181-f008:**
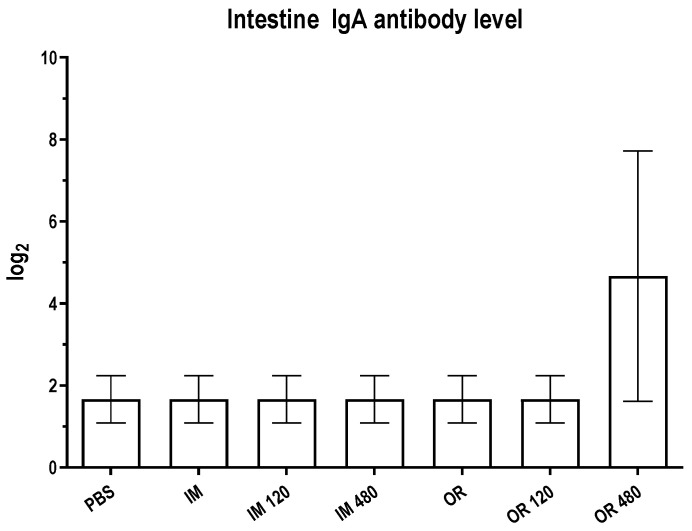
Intestinal IgA antibody level in mice vaccinated with in-house inactivated PEDV and chamomile extracts. Indirect ELISA assays were conducted to measure IgA antibody titer using in-house inactivated PEDV as coating antigen. The statistical analysis used in this study is detailed in Section 2.11.

**Table 1 vetsci-12-00181-t001:** Toxicity test with PEs for mice study.

Group		Dose of Vaccine *	N **
1	Control	PBS	5
2	Chamomile—Low	25 mg/kg bw	5
3	Chamomile—High	50 mg/kg bw	5
4	Mulberry—Low	25 mg/kg bw	5
5	Mulberry—High	50 mg/kg bw	5

* Each mouse received a single dose of 0.1 mL vaccine via intramuscular injection on day 0. ** Number of mice used each group.

**Table 2 vetsci-12-00181-t002:** Vaccination experiments for mice study.

Group		Dose of Vaccine *	N **
A. Intramuscular injection			
1	Control	PBS	5
2	IM	10^5^ TCID_50_ (PEDV)	5
3	IM 120	10^5^ TCID_50_ (PEDV) + 6 mg/kg bw (chamomile)	5
4	IM 480	10^5^ TCID_50_ (PEDV) + 24 mg/kg bw (chamomile)	5
B. Oral administration			
1	Control	PBS	5
2	OR	10^5^ TCID_50_ (PEDV)	5
3	OR 120	10^5^ TCID_50_ (PEDV) + 6 mg/kg bw (chamomile)	5
4	OR 480	10^5^ TCID_50_ (PEDV) + 24 mg/kg bw (chamomile)	5

* For injection, each mouse received two doses (0.1 mL per dose) at an interval of 14 days. For oral administration, each mouse received four doses (0.1 mL per dose) every 7 days. ** Number of mice used each group.

## Data Availability

The original contributions presented in the study are included in the article/Appendix A; further inquiries can be directed to the corresponding author.

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
