# Peer review of "Utilizing NF-κB Signaling in Porcine Epithelial Cells to Identify a Plant-Based Additive for the Development of a Porcine Epidemic Diarrhea Virus Vaccine"

_vetsci, 2025, doi:10.3390/vetsci12020181_

Round 1

Reviewer 1 Report

Comments and Suggestions for Authors

Review of the manuscript

Utilizing NF-κB Signaling in Porcine Epithelial Cells to Identify Plant-Based Adjuvants for Enhanced PEDV Immunization"

Nguyen-Thanh Hoa and colleagues present a manuscript that describes the creation of a porcine epithelial NF-κB reporter cell-line, used to find plant extracts that activate the NF-κB signaling pathway. Out of 224 extracts, 3 showed the induction of NF-κB. To further test these 3 substances for their potential use as adjuvants, in vitro and in vivo toxicity testing were performed, resulting in the selection of chamomile extract as a candidate for an in vivo vaccination study.

The authors used inactivated PEDV in combination with a commercial adjuvant and with or without the addition of chamomile for oral and intramuscular (i.m.) vaccination of mice. The effect of the vaccination was measured by quantifying virus-specific IgG and IgA. The addition of chamomile had no effect for the i.m. vaccination, but showed a tendency to increase antibodies for the oral vaccination. Based on these results, the authors conclude that chamomile extract should be further investigated as a potential adjuvant.

Major Comments:

The authors present a well-written and detailed manuscript describing the screening and in vivo evaluation of a potential adjuvant candidate. The study's rationale is solid, and the authors' dedication to the research is commendable.

However, the obtained results do not fully support the authors' conclusions and claims. While chamomile may induce NF-κB signaling, it showed no effect on intramuscular (i.m.) vaccination, which is not mentioned in the study. Furthermore, even for oral vaccination, high doses of chamomile only resulted in increased antibodies in one experiment. Consequently, the provided data are insufficient to claim that chamomile enhances vaccination.

Therefore, it is also incorrect to refer to chamomile as an adjuvant, as suggested by the title. At best, it may be considered an additive to an adjuvant.

To enhance the quality of the manuscript, I recommend the following:

  1. Change title: The current title suggests that chamomile functions as an adjuvant, but the study's results do not support this claim. A more accurate title would reflect the study's findings.
  2. Improve Figure 2:
    • Add scale bars to all images.
    • Use consistent magnifications for all images.
    • Adjust image brightness and saturation to improve visibility.
    • One picture is oversaturated
  3. Graphical representation:
    • Include the number of repetitions for each data point.
    • Convert bar graphs to dot-plot graphs, similar to Figure 3B.
  4. Graph labels:
    • Clarify what "relative" means in the graphs.
  5. Antibody quantification:
    • Explain why a single point was used to quantify antibodies instead of titers.
    • Consider revising the method to use titers, if applicable.
  6. Negative results:
    • Include a description of the negative results, such as the i.m. vaccination.
  7. Discussion and conclusions:
    • Reevaluate the discussion to address the predictive value of the in vitro model for the in vivo situation.
    • Consider the use of chamomile as an adjuvant in previous studies.
    • Revise the conclusions to accurately reflect the study's findings.

Minor Comments:

Spelling or missing/mixed up numbers

-        line 47, Severe might be several

Summary:

The authors showed a well written manuscript, performed many clever experiments and addressed an important question in the field. While the manuscript has potential, it requires revisions to address the comments made.

Mayor revision is recommended to address the comments made.

Author Response

Reviewer 1’s comment

Authors’ response

Location in the revised manuscript

Nguyen-Thanh Hoa and colleagues present a manuscript that describes the creation of a porcine epithelial NF-κB reporter cell-line, used to find plant extracts that activate the NF-κB signaling pathway. Out of 224 extracts, 3 showed the induction of NF-κB. To further test these 3 substances for their potential use as adjuvants, in vitro and in vivo toxicity testing were performed, resulting in the selection of chamomile extract as a candidate for an in vivo vaccination study.

The authors used inactivated PEDV in combination with a commercial adjuvant and with or without the addition of chamomile for oral and intramuscular (i.m.) vaccination of mice. The effect of the vaccination was measured by quantifying virus-specific IgG and IgA. The addition of chamomile had no effect for the i.m. vaccination, but showed a tendency to increase antibodies for the oral vaccination. Based on these results, the authors conclude that chamomile extract should be further investigated as a potential adjuvant.

Major Comments:

The authors present a well-written and detailed manuscript describing the screening and in vivo evaluation of a potential adjuvant candidate. The study's rationale is solid, and the authors' dedication to the research is commendable.

However, the obtained results do not fully support the authors' conclusions and claims. While chamomile may induce NF-κB signaling, it showed no effect on intramuscular (i.m.) vaccination, which is not mentioned in the study. Furthermore, even for oral vaccination, high doses of chamomile only resulted in increased antibodies in one experiment. Consequently, the provided data are insufficient to claim that chamomile enhances vaccination.

Therefore, it is also incorrect to refer to chamomile as an adjuvant, as suggested by the title. At best, it may be considered an additive to an adjuvant.

To enhance the quality of the manuscript, I recommend the following:

Thank you for your detailed comments. We have revised the manuscript as suggested.

  1. Change title: The current title suggests that chamomile functions as an adjuvant, but the study's results do not support this claim. A more accurate title would reflect the study's findings.

Authors have proposed a new title as below:

“Utilizing NF-κB Signaling in Porcine Epithelial Cells to Identify Plant-Based Additive for the Development of a PEDV Vaccine”

Line 2-3

  1. Improve Figure 2:
    1. Add scale bars to all images.
    2. Use consistent magnifications for all images.
    3. Adjust image brightness and saturation to improve visibility.
    4. One picture is oversaturated

Figure 2 has been revised as suggested.

Line 245

  1. Graphical representation:
    1. Include the number of repetitions for each data point.
    2. Convert bar graphs to dot-plot graphs, similar to Figure 3B.

a. For the dose-finding experiments shown in Figures 3a and 3c, each concentration was tested in triplicate. Specifically, the concentrations of 0, 0.625, 1.25, 2.5, 5, and 10 were evaluated simultaneously to allow for direct comparison.

b. With three replicates for each concentration, our results showed minimal variation, resulting in small error bars. We attempted to present Figures 3a and 3c using dot-plot graphs; however, the large number of concentration points combined with the low variation made the graphs appear cluttered and difficult to interpret. Therefore, we decided to use bar graphs to present the data more clearly.

Line 272-273

  1. Graph labels:
    1. Clarify what "relative" means in the graphs.

We have changed to “fold-change” to clarify our findings.

Figure 3 line 272-273; Figure 4 line 313-314

  1. Antibody quantification:
    1. Explain why a single point was used to quantify antibodies instead of titers.
    2. Consider revising the method to use titers, if applicable.

a. In order to determine even the slightest variances, we aimed to present the ELISA IgA result as a single point, despite the fact that the outcomes from both methods are comparable. However, when samples were diluted in a 1:4 dilution, a negligible increase among the injection groups became apparent.

b. The graph has been revised as suggested.

Figure 8 line 389-390

  1. Negative results:
    1. Include a description of the negative results, such as the i.m. vaccination.

The result sections have been revised as suggested.

Line 367-378; line 383-388

  1. Discussion and conclusions:
    1. Reevaluate the discussion to address the predictive value of the in vitro model for the in vivo situation.
    2. Consider the use of chamomile as an adjuvant in previous studies.
    3. Revise the conclusions to accurately reflect the study's findings.

The discussion and conclusion sections have been rewritten as suggested.

Line 397-472

Minor Comments:

Spelling or missing/mixed up numbers

-        line 47, Severe might be several

The sentence has been corrected.

Line 62

Summary:

The authors showed a well written manuscript, performed many clever experiments and addressed an important question in the field. While the manuscript has potential, it requires revisions to address the comments made.

Mayor revision is recommended to address the comments made.

Reviewer 2 Report

Comments and Suggestions for Authors

After reviewing the research paper entitled "Utilizing NF-κB Signaling in Porcine Epithelial Cells to Identify Plant-Based Adjuvants for Enhanced PEDV Immunization," conducted by Hoa et al. and submitted for evaluation as a potential publication in Veterinary Sciences, I would like to provide the following feedback:

General opinion: This is an interesting study investigating the role of plant extracts (specifically chamomile and mulberry) as adjuvants to enhance the immune response in rats to intramuscular injection or oral challenge with PEDV virus. The manuscript presents a novel and promising approach to identify plant-based adjuvants for PEDV immunization using the PK15-KBR cell line. The study demonstrates the potential of chamomile extract as an immune enhancer, although further validation in larger animal models and additional mechanistic studies would strengthen the conclusions. This research holds promise for advancing the development of effective vaccines for PEDV, and potentially other viral infections in animals, using natural plant compounds as adjuvants.

Changes needed: Overall, the authors did an excellent job with the methodology used in this study. However, in my opinion, there are some shortcomings in the way the statistical data were evaluated. A large portion of the figures indicate that the data analyzed did not meet the assumption of homogeneity of variances (Figures 5, 7, and 8).

Another notable issue is the absence of error bars in Figure 3 (panels A and C) at the 0 concentration. Such data should never be analyzed using a parametric test such as ANOVA.

I recommend that the study data be evaluated using appropriate statistical tests based on the nature of the data distribution. If the authors deem it appropriate, they might consider using a generalized linear mixed model to analyze their data.

The conclusions are incomplete, you need to do a conclusion about your rat study.

Minor Findings:

Line 47: "severe" or "several"?

Line 60: Change "models" to "systems".

Author Response

Reviewer 2’s comment

Authors’ response

Location in the revised manuscript

Comments and Suggestions for Authors

After reviewing the research paper entitled "Utilizing NF-κB Signaling in Porcine Epithelial Cells to Identify Plant-Based Adjuvants for Enhanced PEDV Immunization," conducted by Hoa et al. and submitted for evaluation as a potential publication in Veterinary Sciences, I would like to provide the following feedback:

General opinion: This is an interesting study investigating the role of plant extracts (specifically chamomile and mulberry) as adjuvants to enhance the immune response in rats to intramuscular injection or oral challenge with PEDV virus. The manuscript presents a novel and promising approach to identify plant-based adjuvants for PEDV immunization using the PK15-KBR cell line. The study demonstrates the potential of chamomile extract as an immune enhancer, although further validation in larger animal models and additional mechanistic studies would strengthen the conclusions. This research holds promise for advancing the development of effective vaccines for PEDV, and potentially other viral infections in animals, using natural plant compounds as adjuvants.

Thank you for your detailed comments.

Changes needed: Overall, the authors did an excellent job with the methodology used in this study. However, in my opinion, there are some shortcomings in the way the statistical data were evaluated. A large portion of the figures indicate that the data analyzed did not meet the assumption of homogeneity of variances (Figures 5, 7, and 8).

We have re-evaluated the data using Dunnet's ANOVA and included these findings in our revised manuscript.

The manuscript and graphs have been revised as suggested.

Line

Figure 5 line 331; Figure 7 line 353;Figure 8 line 390

Another notable issue is the absence of error bars in Figure 3 (panels A and C) at the 0 concentration. Such data should never be analyzed using a parametric test such as ANOVA. I recommend that the study data be evaluated using appropriate statistical tests based on the nature of the data distribution. If the authors deem it appropriate, they might consider using a generalized linear mixed model to analyze their data.

We have re-evaluated such data using a student t’s test for Fig 3A-3C and paired t-test for Fig 3B. The manuscript and graphs have been revised as suggested.

Figure 3 line 273; line 282-283

The conclusions are incomplete, you need to do a conclusion about your rat study.

The conclusion section has been rewritten as suggested.

Line 467-473

Minor Findings:

Line 47: "severe" or "several"?

The sentence has been corrected.

Line 62

Line 60: Change "models" to "systems".

The sentence has been corrected.

Line 74

Round 2

Reviewer 1 Report

Comments and Suggestions for Authors

I would like to thank the authors for considering my recommendations. The quality of the manuscript has been significantly improved and now meets the requirements for publication.

Author Response

Comment: I would like to thank the authors for considering my recommendations. The quality of the manuscript has been significantly improved and now meets the requirements for publication.

Responses: Thank you for your feedback and recommendations.

Reviewer 2 Report

Comments and Suggestions for Authors

Thank you for revising your manuscript. I recommend analyzing your data using a generalized linear mixed model, as ANOVA is not appropriate for your dataset. Additionally, please include a supplementary table showing the estimates and 95% confidence intervals (CI) for each of the models.

Author Response

Comment 1: Thank you for revising your manuscript. I recommend analyzing your data using a generalized linear mixed model, as ANOVA is not appropriate for your dataset. Additionally, please include a supplementary table showing the estimates and 95% confidence intervals (CI) for each of the models.

Authors' responses: We have re-analyzed our dataset (figure 3, 5, 7 and 8) using generalized linear mixed models. The manuscript and statistical analysis section have been revised as suggested. The supplementary table has been added.

However, we believe that figure 7 using two-way ANOVA analysis allows for a clearer understanding of the significance between groups and over time. Additionally, we have evaluated this data in GLMM as an additional method, mentioned in supplemental table.

We have referred to the following articles for data analysis, which use ANOVA and t-tests.

  1. El-Murr T, Patel A, Sedlak C, D'Souza-Lobo B. Evaluating dendritic cells as an in vitro screening tool for immunotherapeutic formulations. J Immunol Methods. 2018 Aug;459:55-62. doi: 10.1016/j.jim.2018.05.005. Epub 2018 May 23. PMID: 29800576.
  2. Cetik Yildiz, S., Demir, C., Cengiz, M. et al. The protection afforded by kefir against cyclophosphamide induced testicular toxicity in rats by oxidant antioxidant and histopathological evaluations. Sci Rep 14, 18463 (2024). https://doi.org/10.1038/s41598-024-67982-y
  3. 3. Yu, J., Sreenivasan, C., Uprety, T. et al. Piglet immunization with a spike subunit vaccine enhances disease by porcine epidemic diarrhea virus. npj Vaccines 6, 22 (2021). https://doi.org/10.1038/s41541-021-00283-x

Location in the revised manuscript: Line 237-242; Figure 3 line 274; line 281-282; line 318-319; Figure 5 line 328; line 330-331; line 354-355; line 385-386

Round 3

Reviewer 2 Report

Comments and Suggestions for Authors

Please address my concerns. This is the second round of review, and I noticed that my previous feedback has not been addressed. While your research work is excellent, I believe the statistical analysis needs improvement. I recommend using the free software JASP for analyzing your data, as it will help you present the figures with a more appropriate model.

Author Response

Comment: Please address my concerns. This is the second round of review, and I noticed that my previous feedback has not been addressed. While your research work is excellent, I believe the statistical analysis needs improvement. I recommend using the free software JASP for analyzing your data, as it will help you present the figures with a more appropriate model.

Responses: 

Thank you for your insightful feedback and for bringing to our attention that the statistical analysis in our manuscript requires improvement. To address your concerns, we have reanalyzed the data using GLMMs test; however, we failed to get the reliable outcomes, which may be due to our research design and sample size. For the dose-finding experiments shown in this study, each data was tested in triplicate. Specifically, the continuous variables (concentrations) were evaluated simultaneously to allow for direct comparison. After consulting with a professional statistician to validate our analysis and interpretation of the results, we have revised the figures and added supplemental tables to present the data more clearly and accurately, as follows:

Figure 3 (panel A and B) has been analyzed using non-parametric Kruskal-Wallis Test. Figure 3 (panel C) has been analyzed using independent samples t-test for comparing two groups (detailed analysis is provided in Supplemental Data Figure 3). 

First, the data from Figure 5 was analyzed to determine the differences in concentrations, groups and interaction term. Then, each treatment group was analyzed using the non-parametric Kruskal-Wallis test. An independent sample t-test was performed to determine the difference between the positive control (H2O2 group) and the negative control (DMSO) (detailed analysis has been provided in Supplemental Data Figure 5).

Data from figure 7 was first analyzed using two-way ANOVA to identify statistically significant differences between groups and over time, and then re-analyzed using GLMMs for an alternative test (detailed analysis has been provided in Supplemental Data Figure 7).  

Figure 8 has been analyzed using non-parametric Kruskal-Wallis Test (detailed analysis has been provided in Supplemental Data Figure 8)

Moreover, we noticed that Prism provides clearer graphs with statistical significance, which we couldn't achieve with JASP. Therefore, we have created our graphs using Prism software.

We hope that these explanations will address your concerns and improve the rigor and clarity of our statistical analysis. Thank you again for your valuable input. We look forward to your assessment of our revised manuscript.